# HINDSIGHT-DICE: STABLE CREDIT ASSIGNMENT FOR DEEP REINFORCEMENT LEARNING

## ABSTRACT

Oftentimes, environments for sequential decision-making problems can be quite sparse in the provision of evaluative feedback to guide reinforcement-learning agents. In the extreme case, long trajectories of behavior are merely punctuated with a single terminal feedback signal, leading to a significant temporal delay between the observation of a non-trivial reward and the individual steps of behavior culpable for achieving said reward. Coping with such a credit assignment challenge is one of the hallmark characteristics of reinforcement learning. While prior work has introduced the concept of hindsight distributions (Harutyunyan et al., 2019) to develop a principled method for reweighting on-policy data according to impact on achieving the observed trajectory return, we show that these methods experience instabilities which cause inefficient learning in complex environments. In this work, we adapt existing importance-sampling ratio estimation techniques for off-policy policy evaluation to drastically improve the stability and efficiency of these so-called hindsight distribution methods. Our hindsight distribution correction facilitates stable, efficient learning across a broad range of environments where credit assignment plagues baseline methods.

## 1 INTRODUCTION

Reinforcement learning is the classic paradigm for addressing sequential decision-making problems (Sutton & Barto, 1998). Naturally, while inheriting the fundamental challenge of generalization across novel states and actions from supervised learning, general-purpose reinforcement-learning agents must also contend with the additional challenges of exploration and credit assignment. While much initial progress in the field was driven largely by innovative machinery for tackling credit assignment (Sutton, 1984; 1988; Singh & Sutton, 1996) alongside simple exploration heuristics ($\varepsilon$-greedy exploration, for example), recent years have seen a reversal with the bulk of attention focused on a broad array of exploration methods, spanning additional heuristics as well as more principled approaches (Tang et al., 2017; Pathak et al., 2017; Ecoffet et al., 2021), and relatively little consideration given to issues of credit assignment. This lack of interest in solution concepts, however, has not stopped the proliferation of reinforcement learning into novel application areas characterized by long problem horizons and sparse reward signals. Apart from recent works such as AlphaStar (Vinyals et al., 2019) and DeepNash for Stratego (Perolat et al., 2022), the current reinforcement learning from human feedback (RLHF) paradigm (Lambert et al., 2022) is now a widely popularized example of an environment that operates in perhaps the harshest setting where a single feedback signal is only obtained after the completion of a long trajectory.

This combination of long horizons with delayed rewards most directly impacts an agent's ability to efficiently perform credit assignment (Minsky, 1961; Sutton, 1984; 1988) and thereby worsens the overall sample complexity needed to synthesize an optimal policy. Consequently, it is perhaps the credit assignment challenge that poses the greatest impediment to sample-efficient reinforcement learning. While theoretical results encapsulating settings like that of the RLHF paradigm are nascent (Chatterji et al., 2021; Pacchiano et al., 2023), practical and demonstrably-scalable mechanisms for efficient credit assignment are largely absent from the literature.

With this reality in mind, our work extends the relatively recent solution concept of hindsight distributions as introduced by Harutyunyan et al. (2019), which allow an agent to reason counterfactually about past decisions at particular states in light of observed future outcomes. While the natural

course of action for algorithm design based on the provision of such a hindsight distribution is to reweight the on-policy data collected for agent updates, we highlight a key issue where the resulting importance-sampling ratio is highly unstable and compromises the quality of learning. Even when ratio clipping is applied, as one often does in off-policy policy-gradient methods (Schulman et al., 2017), such a heuristic lends stability at the heavy cost of incredibly dampened learning speed.

To remedy this instability, we turn to prior work tackling the orthogonal problem of off-policy policy evaluation (OPE) (Precup et al., 2000; Jiang & Li, 2016; Thomas & Brunskill, 2016) wherein the ratio between state visitation probabilities of a behavior policy and target policy has emerged over recent years as a key quantity of interest for yielding principled, practical OPE algorithms (Hallak & Mannor, 2017; Liu et al., 2018; Gelada & Bellemare, 2019; Liu et al., 2019; 2020; Nachum et al., 2019; Uehara et al., 2020; Zhang et al., 2020a;b; Dai et al., 2020) whose variance does not suffer exponentially in the problem horizon, as it would with a naive application of importance sampling. Inspired by the potential for technical progress in this separate arena to be repurposed and operationalized towards achieving more efficient credit assignment techniques, we select one such approach from the OPE literature known as Dual stationary DIstribution Correction Estimation (DualDICE) (Nachum et al., 2019) and adapt the importance ratio approximation technique for hindsight distributions. Our resulting **H**indsight **DI**stribution **C**orrection **E**stimation (**H-DICE**) approach empowers policy-gradient methods (Schulman et al., 2017) to empirically avoid the same catastrophic instability that arises by naively computing the importance ratio between the current policy and hindsight distribution. We empirically demonstrate over three different and challenging benchmarks tasks that our method outperforms baseline approaches, which either do not incorporate any deliberate mechanism for tackling credit assignment or do so in a naive fashion, by a large margin both in terms of the achieved final rewards and the speed of convergence.

The paper proceeds as follows: in Section 2 we detail our problem formulation as well as background information on credit assignment with hindsight distributions. In Section 3, we outline our approach before presenting a detailed empirical evaluation and discussion of results in Section 4. Due to space constraints, we relegate an overview of related work, mathematical expansions, and ablation studies to the Appendix.

## 2 Preliminaries & Background

In this section, we begin by specifying our problem formulation before providing brief background information on the credit assignment problem broadly as well as the specific use of hindsight distributions to address the challenge.

For any natural number $N \in \mathbb{N}$, we denote the index set as $[N] \triangleq \{1, 2, \ldots, N\}$. For any set $\mathcal{X}$, we denote the set of all probability distributions with support on $\mathcal{X}$ as $\Delta(\mathcal{X})$. For another arbitrary set $\mathcal{Y}$, we denote the class of all functions mapping from $\mathcal{X}$ to $\mathcal{Y}$ as $\{\mathcal{X} \to \mathcal{Y}\}$.

### 2.1 Problem Formulation

We formulate a sequential decision-making problem as an infinite-horizon, discounted Markov Decision Process (MDP) (Bellman, 1957; Puterman, 1994) defined by $\mathcal{M} = \langle \mathcal{S}, \mathcal{A}, \mathcal{R}, \mathcal{T}, \mu, \gamma \rangle$. Here $\mathcal{S}$ denotes a set of states, $\mathcal{A}$ is a set of actions, $\mathcal{R} : \mathcal{S} \times \mathcal{A} \to \mathbb{R}$ is a deterministic reward function providing evaluative feedback signals to the agent, $\mathcal{T} : \mathcal{S} \times \mathcal{A} \to \Delta(\mathcal{S})$ is a transition function prescribing distributions over next states, $\mu \in \Delta(\mathcal{S})$ is an initial state distribution, and $\gamma \in [0, 1)$ is the discount factor communicating a preference for near-term versus long-term rewards. Beginning with an initial state $s_0 \sim \mu$, for each timestep $t \in \mathbb{N}$, the agent observes the current state $s_t \in \mathcal{S}$, selects action $a_t \sim \pi(\cdot \mid s_t) \in \mathcal{A}$, receives a reward $r_t = \mathcal{R}(s_t, a_t)$, and transitions to the next state $s_{t+1} \sim \mathcal{T}(\cdot \mid s_t, a_t) \in \mathcal{S}$.

Our reinforcement-learning approach is grounded in policy search (Williams, 1992; Sutton et al., 1999a; Konda & Tsitsiklis, 1999; Mnih et al., 2016; Schulman et al., 2017) wherein the agent's behavior is represented by a stationary, stochastic policy $\pi_\theta : \mathcal{S} \to \Delta(\mathcal{A})$, parameterized by $\theta \in \Theta \subset \mathbb{R}^d$, that encodes a pattern of behavior mapping individual states to distributions over possible actions. The overall performance of $\pi_\theta$ in any MDP $\mathcal{M}$ when starting at state $s \in \mathcal{S}$ and taking action $a \in \mathcal{A}$ is assessed by its associated action-value function $Q^{\pi_\theta}(s, a) =$

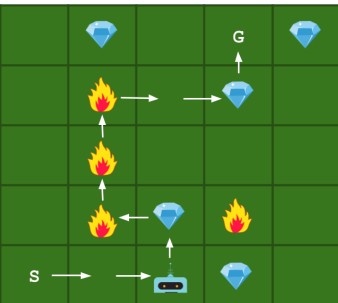
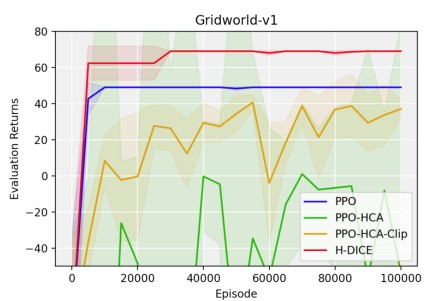

Figure 1: *Left:* GridWorld-v1 environment, described in Sec. 4.1, and an example trajectory (denoted by arrows) taken by the agent (denoted by the robot). The agent must collect diamonds while avoiding fire, and only receives a meaningful reward signal upon episode termination. *Right:* Learning curves of our proposed method, H-DICE, and baseline methods (PPO, PPO-HCA, and PPO-HCA-Clip) on Gridworld-v1. H-DICE achieves the maximum reward quickly and stably compared to other methods. PPO learns a suboptimal policy, likely because it does not explicitly model credit-assignment. While using unclipped HCA ratios to compute advantages (PPO-HCA) results in highly unstable performance, clipping the ratios (PPO-HCA-Clip) improves results slightly.

$\mathbb{E}\left[\sum_{t=0}^{\infty} \gamma^t \mathcal{R}(s_t, a_t) \mid s_0 = s, a_0 = a\right]$, where the expectation integrates over randomness in the action selections and transition dynamics. Taking the corresponding value function as $V^{\pi_\theta}(s) = \mathbb{E}_{a \sim \pi_\theta(\cdot|s)}[Q^{\pi_\theta}(s, a)]$ and letting $\Pi_\Theta \triangleq \{\pi_\theta \mid \theta \in \Theta\} \subset \{\mathcal{S} \to \Delta(\mathcal{A})\}$ denote the policy class parameterized by $\Theta$ (a subset of all stochastic policies), we define the optimal policy $\pi^\star$ as achieving supremal value with respect to this class $V^\star(s) = \sup_{\pi \in \Pi_\Theta} V^\pi(s) = \max_{a^\star \in \mathcal{A}} Q^\star(s, a^\star)$ and $Q^\star(s, a) = \sup_{\pi \in \Pi_\Theta} Q^\pi(s, a)$ for all $s \in \mathcal{S}$ and $a \in \mathcal{A}$. We may examine the visitation of a policy $\pi_\theta$ through its discounted, stationary state distribution: $d^{\pi_\theta}(s) = (1 - \gamma) \sum_{t=0}^{\infty} \gamma^t \mathbb{P}^\pi(s_t = s)$, where $\mathbb{P}^\pi(s_t = \cdot) \in \Delta(\mathcal{S})$ denotes the distribution over states visited by $\pi_\theta$ at each timestep $t$. Finally, in alignment with prior work (Bellemare et al., 2017; Harutyunyan et al., 2019), we use $Z(\tau)$ to denote the random variable representing the return of a trajectory $\tau$.

## 2.2 A Motivating Example for Credit Assignment

Consider the gridworld setting depicted in Figure 1, wherein an agent accrues positive rewards for collecting diamonds and negative rewards for moving through fire. Importantly, however, the agent only receives a nonzero reward signal, which is the sum of the rewards accrued in each individual transition, at the end of a trajectory when it enters the $G$ square and receives no rewards for the intermediary steps.

In the example trajectory illustrated in Figure 1, the agent first collects a diamond and then transitions through three fire squares before acquiring a diamond and reaching $G$, upon which it receives a large negative reward (due to the three steps into the fire). While a typical policy-gradient method would place "blame" for this poor outcome on *each* action in the trajectory and would correspondingly reduce the probability that these actions are taken by the policy, the actions responsible for collecting the diamonds were optimal and these actions should be made more likely despite the large negative reward. Effective credit assignment aims to exactly reach this type of conclusion; not all actions are equally responsible for a future outcome, and determining which actions are responsible in yielding a particular end result can enable more efficient policy learning and improved policy performance.

## 2.3 Hindsight Credit Assignment

More generally, the credit assignment problem revolves around a given behavior policy of an agent and asks *what is the impact of being in a particular state and executing a particular action on observed future outcomes?* (Harutyunyan et al., 2019; Arumugam et al., 2021). Typically, the future

outcome of interest is the full return $Z(\tau)$ associated with a complete trajectory $\tau$, although one could also examine intermediate returns observed from any incident states or state-action pairs.

While classic approaches to addressing credit assignment would distribute credit based on temporal recency to interesting events or outcomes (Klopf, 1972; Sutton, 1984; Singh & Sutton, 1996), a more recent and cogent mechanism has emerged through the use of a *hindsight distribution* (Harutyunyan et al., 2019). For the purposes of this work, a hindsight distribution $h_\omega^{\pi_\theta} : \mathcal{S} \times \mathbb{R} \to \Delta(\mathcal{A})$ induced by a behavior policy $\pi_\theta$ and parameterized by $\omega \in \Omega \subset \mathbb{R}^m$ is a return-conditioned distribution over actions taken by policy $\pi_\theta$ from a given state and in light of an observed trajectory return. The utility of the hindsight distribution becomes apparent from the following identity for the advantage function (Baird III, 1993) computed by policy-gradient (Mnih et al., 2016; Schulman et al., 2017; 2018) methods:

$$A^{\pi_\theta}(s,a) = Q^{\pi_\theta}(s,a) - V^{\pi_\theta}(s) = \mathbb{E}\left[\left(1 - \frac{\pi_\theta(a \mid s)}{h_\omega^{\pi_\theta}(a \mid s, Z(\tau))}\right) Z(\tau)\right]. \tag{1}$$

If, in hindsight, observing a return $Z(\tau)$ makes a particular action less likely, then the *hindsight ratio* $\frac{\pi_\theta(a|s)}{h_\omega^{\pi_\theta}(a|s,Z(\tau))} > 1$ and detracts from the advantage of the action; in contrast, an action that makes the return $Z(\tau)$ more plausible in hindsight results in an increased advantage signal as $\frac{\pi_\theta(a|s)}{h_\omega^{\pi_\theta}(a|s,Z(\tau))} < 1$. More generally, as the value of the hindsight ratio increases, this update equation reduces the culpability of the given action for the observed outcome. In the next section, we discuss the practical challenges that emerge when estimating this hindsight ratio so as to integrate this credit assignment information within a policy-gradient method.

## 3 APPROACH

We begin by examining the natural, straightforward approach to hindsight ratio estimation and highlight stability issues that arise from using such an estimator. We then proceed to introduce one alternative avenue for hindsight ratio estimation inspired by prior work in OPE. Finally, we conclude by introducing our H-DICE approach, which leverages this ratio estimator for demonstrably stable credit assignment.

### 3.1 DIRECT DENSITY RATIO INSTABILITY

As detailed in Section 2.3, the HCA framework derives an expression for advantage function $A^{\pi_\theta}(s,a)$ of a policy $\pi_\theta$, which, through the return-conditioned hindsight distribution $h^{\pi_\theta}$, explicitly assigns the credit each action receives for achieving a particular return from the current state. Given on-policy trajectory data $\mathcal{D}^{\pi_\theta}$, a direct way to compute this advantage would be to first fit a hindsight distribution $h_\omega^{\pi_\theta}$ to the data via supervised learning and then compute the hindsight ratio as $\frac{\pi_\theta(a|s)}{h_\omega^{\pi_\theta}(a|s,Z(\tau))}$, for each state-action pair $(s,a)$ in a trajectory $\tau$. $A^\pi(s,a)$ can subsequently be computed using expression 1, as done in the approach of Harutyunyan et al. (2019).

However, direct estimation of density ratios can be highly unstable and, as shown in the right of Figure 1 and Section 4.5, the hindsight ratio is no exception. Using Proximal Policy Optimization (PPO) (Schulman et al., 2017) as a base policy-gradient algorithm, we empirically observe that values the ratio takes through the course of training vary greatly, hindering policy learning (see curve PPO-HCA in 1). Clipping the ratios to be between zero and one, a common way to stabilize density ratios (Schulman et al., 2017), improves the stability of learning but still results in suboptimal performance, as seen by the results of PPO-HCA-Clip. Thus, the question remains: how can we compute the hindsight ratio in a *stable* manner?

### 3.2 HINDSIGHT DISTRIBUTION CORRECTION ESTIMATION

In this work, we offer one candidate answer to the above question by taking inspiration from the off-policy policy evaluation (OPE) literature, where recent methods have been developed to accurately and soundly compute the importance sampling ratio between two state-action distributions. Specifically, we leverage the optimization problem posed by Nachum et al. (2019) in the DualDICE method.

Our H-DICE method for stable computation of the hindsight ratio $\frac{\pi_\theta(a|s)}{h_\omega^{\pi_\theta}(a|s,Z(\tau))}$ adapts their optimization problem to our setting by focusing on the following optimization over a function $\phi : \mathcal{S} \times \mathcal{A} \times Z \to \mathcal{C}$ with bounded range $\mathcal{C} \triangleq [0, C]$, for some numerical constant $C \in \mathbb{R}_{\geq 0}$:

$$\min_{\phi: S \times A \times \mathbb{R} \to \mathcal{C}} \frac{1}{2} \mathbb{E}_{(s,a,z) \sim \mathcal{D}^{h^{\pi_\theta}_\omega}} [\phi(s,a,z)^2] - \mathbb{E}_{\substack{(s,a) \sim d^{\pi_\theta} \\ z \sim \psi(z)}} [\phi(s,a,z)] \tag{2}$$

where $\mathcal{D}^{h^{\pi_\theta}}$ denotes an empirical state-action-return distribution formed by drawing states under the $d^{\pi_\theta}$ visitation of $\pi_\theta$, returns of $\pi_\theta$ observed from state $s$ according to $\chi^{\pi_\theta}(\cdot|s)$, and actions from the hindsight distribution $h_\omega^{\pi_\theta}(\cdot \mid s, z)$. $\psi(z)$ is an arbitrary distribution over the returns that, in our experiments, we model as a uniform distribution over all possible returns in the environment, which results in simplified hindsight ratio calculation. In the main results presented in Section 4, we set the range threshold $C \in \mathbb{R}_{\geq 0}$ for $\phi$ to be 1; however, as shown in Appendix D, the choice of $C$ is largely inconsequential and numerous values result in similar performance.

While Appendix B elaborates in more detail, we briefly mention here that the solution to this optimization, $\phi^\star$, is given by:

$$\phi^\star(s, a, z) = \frac{\pi_\theta(a|s)}{\chi^{\pi_\theta}(z|s) h_\omega^{\pi_\theta}(a|s,z)}.$$

This allows us to arrive at the following expression for the hindsight ratio of interest:

$$\frac{\pi_\theta(a|s)}{h_\omega^{\pi_\theta}(a|s,z)} = \phi^\star(s,a,z) \chi^{\pi_\theta}(z|s) \tag{3}$$

Importantly, the right hand side of Equation 3 can be estimated from on-policy data $\mathcal{D}^{\pi_\theta}$ collected under $\pi_\theta$. This involves learning the following models, which we instantiate as neural networks:

1. **Return predictor** $\chi_\eta^{\pi_\theta}$: We learn the distribution over returns incurred by policy $\pi_\theta$ given a state $s \in \mathcal{S}$ via supervised learning using $\mathcal{D}^{\pi_\theta}$. Note that our approach can be seen as a Monte-Carlo estimation of the return distribution and $\chi_\eta^{\pi_\theta}$ could also be learned via distributional RL (Bellemare et al., 2017), an avenue we leave to future work.

2. **Hindsight distribution** $h_\omega^{\pi_\theta}$: Just as the approach taken by Harutyunyan et al. (2019), we learn a hindsight distribution via supervised learning using on-policy data $\mathcal{D}^{\pi_\theta}$.

3. **Hindsight DICE model** $\phi_\nu$: Lastly, our estimate of $\phi^\star$ in Equation 3 is given by $\phi_\nu$ optimized via Equation 2. Naturally, this requires $\mathcal{D}^{\pi_\theta}$ along with $\chi_\eta^{\pi_\theta}$ and $h_\omega^{\pi_\theta}$.

As all the aforementioned models used to compute hindsight ratios are (either implicitly or explicitly) conditioned on the current policy, $\pi_\theta$, we must reset the weights of the corresponding function approximators after each policy update and before collecting fresh data from the latest policy. In our experiments, since these models are implemented as small fully-connected networks, we find that this does not adversely affect wall-clock time or the computational efficiency of our algorithm.

Like DualDICE, our approach for estimating the hindsight ratio inherits an avoidance of explicitly computing importance-sampling weights, thereby alleviating the instability that arises from direct density ratio estimation observed in Section 3.1. The full algorithm for policy learning using H-DICE is detailed in Algorithm 1. All the models are trained together within the same loop and there is no required pre-training. Note that although H-DICE and all baseline methods use PPO (Schulman et al., 2017) as the central policy learning algorithm, our approach is agnostic to this choice and compatible with any policy-gradient method. Notably and in contrast to many standard PPO implementations which learn a value-function alongside a policy to compute advantages, H-DICE does not learn a value-function to compute advantages as a consequence of Equation 1.

---

**Algorithm 1** Hindsight Distribution Correction Estimation

---

Initialize $\pi_{\theta_1}$
**for** iteration $i = 1 \ldots N$ **do**
    Initialize $h_\omega^{\pi_{\theta_i}}$, $\phi_\nu$, and $\chi_\eta^{\pi_{\theta_i}}$
    Collect a batch of on-policy trajectories $\mathcal{D}^{\pi_{\theta_i}}$ with $\pi_{\theta_i}$
    Train $\chi_\eta^{\pi_{\theta_i}}$ and $h_\omega^{\pi_{\theta_i}}$ by supervised learning with $\mathcal{D}^{\pi_{\theta_i}}$
    Train H-DICE $\phi_\nu$ by Equation 2 using $\chi_\eta^{\pi_{\theta_i}}$ and $h_\omega^{\pi_{\theta_i}}$
    Obtain $\theta_{i+1}$ via PPO update on $\pi_{\theta_i}$ with advantages in Equation 1
**end for**

---

## 4 EXPERIMENTS

In this section, we evaluate H-DICE on complex discrete and continuous control tasks to compare it against baselines and to investigate the learned hindsight ratios. For the sake of brevity, we defer experiments on the impact of the constant $C$, the use different distributions for $\psi$, exploration of varying update schedules for the auxiliary models, and using the H-DICE advantage estimator on dense reward settings to the Appendix. We first provide the details of our evaluation domains and chosen baselines.

### 4.1 DOMAINS

We evaluate methods on the following domains:

**GridWorld:** The GridWorld environments, shown in Figures 1 and 8, are environments we develop which pose a difficult credit-assignment challenge. In each square, the agent can take actions representing the cardinal directions. Each square in the grid may contain a diamond or a fire; transitioning into a diamond square gives the agent a reward of +20, while running into a fire square gives the agent a reward of -100. Moreover, each step the agent takes incurs a reward of -1. An episode ends either when the agent transitions into the goal square or when the episode length passes a predetermined threshold. We create two such grid world settings: GridWorld-v1, a smaller grid with a maximum episode length of 50, and GridWorld-v2, a larger grid with a maximum episode length of 100. An optimal agent will accumulate diamonds as quickly as possible while avoiding fire, and eventually navigate to the goal square to finish the episode.

The credit-assignment challenge in this domain arises from the fact that all rewards are delayed until the very end of an episode; in other words, the sum of the rewards received for each individual transition is given as a single cumulative reward at the end of the episode, and all intermediate rewards given to the agent are zero. We note that delaying the reward in this manner violates the Markov property of the MDP reward function and, technically, would be better modeled as a partially-observable MDP (POMDP) (Kaelbling et al., 1998) where rewards are a function of a latent underlying state not observable to the agent. However, the optimal policy for this particular POMDP (and future delayed-reward settings we will introduce) can still be learned purely as a function of the partial observation, obviating the need for considering the underlying latent state or redefining states around the full trajectory. We also note that this approach of delaying rewards in an MDP while keeping the rest of the structure the same has been studied extensively in prior work (Arjona-Medina et al., 2019; Hung et al., 2019; Ren et al., 2022; Gangwani et al., 2020).

This delayed reward setting forces a good agent to disentangle the fact that collecting diamonds results in positive rewards while moving through fire results in bad rewards by assigning credit to individual actions within an episode. As we demonstrate in the following sections, this is a challenging task for algorithms which do not perform explicit credit assignment.

**LunarLander:** To evaluate methods on a more complex discrete action environment, we turn to the LunarLander-v2 benchmark from OpenAI Gym (Brockman et al., 2016), in which an agent attempts to successfully land a spaceship within a specified zone. Again, to impose a credit-assignment challenge in this setting, the environment is modified such that rewards are delayed to the end of the

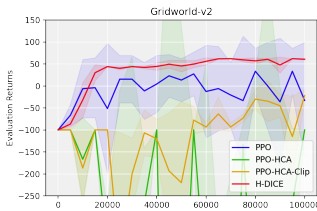 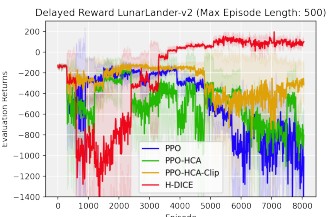 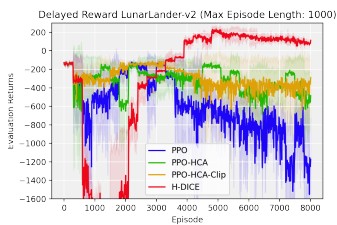

Figure 2: Training curves of H-DICE and baseline methods in Gridworld-v2 (left), and delayed reward LunarLander-v2 with maximum episode lengths of 500 (middle) and 1000 (right). The LunarLander-v2 environment is considered "solved" when returns surpass 200. All environments utilize a discrete action space. H-DICE outperforms baseline methods in all three environments.

episode. We additionally evaluate the agent with maximum episode lengths of 500 and the default length of 1000. The environment is considered "solved" when the agent achieves a return of 200.

**MuJoCo Gym Suite:** Lastly, we benchmark methods on complex continuous control settings using four environments from the MuJoCo suite (Todorov et al., 2012): HalfCheetah, Humanoid, Walker2d, and Swimmer. Again, all rewards are delayed to episode termination and, additionally, episodes in all environments are truncated in length to a maximum of 100 steps to keep learning tractable. We also evaluate all methods in HalfCheetah with a maximum episode length of 50, as part of an ablation study.

## 4.2 BASELINES

We compare H-DICE to three baseline approaches:

**Vanilla PPO:** Our first baseline is PPO with GAE-advantage calculation (Schulman et al., 2017; 2018), a method which does not perform explicit credit-assignment.

**PPO-HCA:** This baseline calculates the advantage as proposed by return-conditioned HCA (Harutyunyan et al., 2019) instead of the typical Generalized Advantage Estimate (Schulman et al., 2018) used in PPO, but computes the hindsight ratio directly by simply dividing the policy and hindsight distribution likelihoods. As shown in the results, this method of computing hindsight ratios is highly unstable and results in poor policy performance.

**PPO-HCA-Clip:** This baseline is identical to PPO-HCA, but attempts to stabilize the hindsight ratio (which is still computed by dividing the policy and hindsight distribution ratios) by clipping the ratio to be between zero and one. We choose these bounds since we find smaller clip ranges improves performance and since the density ratios in H-DICE are between zero and one by virtue of the fact that, by default when $C = 1$, the hindsight DICE model and return predictor models output values between zero and one.

## 4.3 EXPERIMENTAL SETUP

We tune H-DICE and each baseline method via a limited grid search over critical hyperparameters. Specifically, for each domain, we search over the entropy bonus coefficient and the batch size collected before each policy update for each method. For HCA methods and H-DICE, we additionally tune the number of epochs of supervised learning for the hindsight model, return predictor, and DICE model. Lastly, for the vanilla PPO baseline, we also tune the coefficient on the value-function loss. We refer the reader to Appendix G for a detailed account of hyperparameters used. All results are averaged over three seeds; the solid lines in the plots correspond to the mean return and the shaded region to one standard deviation from the mean.

## 4.4 HOW DOES H-DICE PERFORM COMPARED TO BASELINES ON COMPLEX CONTINUOUS CONTROL AND DISCRETE CONTROL TASKS IN THE DELAYED REWARD SETTING

Figures 1, 2 and 3 display learning curves of H-DICE and baseline algorithms in a variety of different discrete and continuous control environments. The vertical axis denotes the returns achieved by

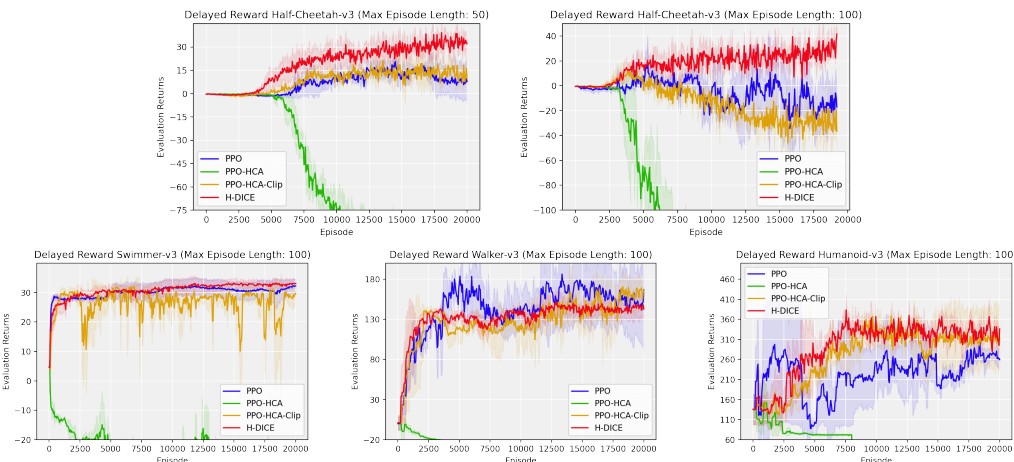

Figure 3: Training curves of H-DICE and baseline methods in Half-Cheetah with a maximum episode length of 50 and 100, and in other MuJoCo environments with a maximum episode length of 100. H-DICE outperforms baseline methods in the more difficult Half-Cheetah and Humanoid environments, and performs similarly to PPO and PPO-HCA-Clip on Swimmer and Walker.

the agent averaged over 10 evaluation episodes whereas the horizontal axis denotes the number of episodes elapsed since the beginning of training.

H-DICE achieves higher final returns and converges faster than baseline methods in the vast majority of scenarios. PPO notably struggles to achieve high returns in many of these domains and is typically outperformed by H-DICE in final returns and learning efficiency. The exception to this is in the Swimmer and Walker environments, where PPO, PPO-HCA-Clip, and H-DICE perform similarly. This demonstrates the necessity for explicit credit-assignment in settings where the lack of immediate reward feedback requires reasoning about actions which were responsible for generating the final return. As illustrated by the poor results of PPO-HCA across domains, computing hindsight ratios directly by dividing the policy and hindsight distribution likelihoods results in instability issues and leads to suboptimal performance; the variance across seeds is extremely high, and the achieved returns oscillate drastically particularly in the GridWorld and LunarLander settings. Stabilizing the hindsight ratios via clipping, the strategy used by PPO-HCA-Clip, alleviates this instability in most settings, as evidenced by both the improvement in returns and the reduction in variance as compared to PPO-HCA. As it still explicitly models credit-assignment, PPO-HCA-Clip improves upon PPO in several settings but is clearly outperformed by H-DICE across most environments, demonstrating the need for more sophisticated methods for stable hindsight ratio computation.

We also note several interesting trends in the relationship between episode length and method performance. As illustrated by the final achieved rewards in the LunarLander and Half-Cheetah plots in Figures 2 and 3 respectively, the difference in performance between H-DICE and baseline methods (particularly PPO) is more pronounced when the maximum episode length is longer. This may be explained by the fact that credit assignment becomes increasingly important over longer horizons as the effects of earlier actions can manifest much later in the episode as well as due to the temporal distance between each action execution and received reward is greater.

### 4.5 DO THE HINDSIGHT RATIOS LEARNED BY H-DICE ENCODE MEANINGFUL CREDIT ASSIGNMENT TRENDS?

To answer this question, we take a fully trained H-DICE model on GridWorld-v1 and analyze the learned $\pi_\theta$ and $h_{\pi_\theta}$ values at a critical state in a trajectory – namely, when the agent is in the position shown in Figure 1. In this position, the agent can take the RIGHT action to collect a diamond and go on to get large positive reward but it may also run into fires as shown in the trajectory in Figure 1 and accrue a large negative reward.

In the table below, we show values corresponding to the RIGHT action taken by the agent at this state ($s$), for two hypothetical, yet representative, achieved final returns ($z$) in this environment. The probability of taking that action under policy $\pi_\theta$ at this state and the hindsight probability, $h_\omega^{\pi_\theta}(a \mid s, z)$, of choosing this action at this state given the final return and the current policy are also shown. From these values, we also show the naive ratio calculation of $\frac{\pi_\theta(a|s)}{h_\omega^{\pi_\theta}(a|s,z)}$ and the same ratio obtained via H-DICE using Equation 3.

Table 1: Table showing the learned values for action RIGHT from H-DICE at the end of training in GridWorld-v1 and the corresponding ratios obtained from naive division and H-DICE.

| Return (z) | $\pi_\theta(a \mid s)$ | $h_\omega^{\pi_\theta}(a \mid s, z)$ | Direct Ratio | H-DICE Ratio |
|---|---|---|---|---|
| -100 | 0.785 | 0.290 | 2.706 | 0.168 |
| 70 | 0.785 | 0.465 | 1.687 | 0.012 |

From column 3 of the table, we can see that the learned hindsight distribution model indicates that the probability of taking the RIGHT action and collecting a diamond is much higher for a large positive return than it is for a large negative return.

The final two columns of the table show the direct $\frac{\pi_\theta(a|s)}{h_\omega^{\pi_\theta}(a|s,z)}$ ratio values which are computed by simply dividing the two required probabilities, and the approximated ratio obtained via H-DICE using Equation 3. We observe that the direct ratio, has much larger values, therefore leading to larger variance and instability in learning even in a simple environment such as GridWorld-v1. On the other hand, ratios computed using H-DICE are much smaller and hence have lower variance, while still mirroring the trends of the direct ratio. We posit that this ability to capture the direct ratio's trends while reducing variance is a core reason for the improved performance of H-DICE compared to PPO-HCA. However, there are times when inaccuracies in auxiliary models introduce bias into the ratio calculation and prevent H-DICE from reflecting the trends of the direct ratio. Nevertheless, as backed by the results in Section 4, H-DICE succesfully trades off bias for variance and this is crucial to achieving strong empirical results in challenging credit-assignment settings. Lastly, while simple settings such as GridWorld-v1 enable us to interpret model outputs at the level of individual state-action pairs, we suspect that these issues are only exacerbated in more complex environments, further underscoring the importance of credit assignment in deep RL.

## 5  CONCLUSION

In this work, we presented H-DICE, a novel algorithm for *stable* credit assignment in deep reinforcement learning. Building on the Hindsight Credit Assignment (HCA) framework introduced by Harutyunyan et al. (2019) and noting the instabilities that arise when directly estimating hindsight density ratios, we take inspiration from methods in the off-policy policy evaluation literature (Nachum et al., 2019) to develop a method which computes the hindsight ratio in a stable fashion. We showed that, in challenging benchmarks with delayed rewards, algorithms that do not assign credit or do so in a naive fashion struggle to perform well while H-DICE converges to higher returns much faster and more consistently than baseline methods.

Our work utilizes a formulation of HCA in which the hindsight distribution is conditioned on future outcome information in the form of returns. However, it is likely that H-DICE will struggle in sparse-reward settings where there is a lack of diversity in observed returns. To address this challenge in future work, we hope to extend this work to state-conditioned HCA, a formulation in which the hindsight distribution is conditioned on future states in a trajectory rather than the return. We expect this variant to fare better in sparse-reward, language conditioned and goal-reaching settings.

While we conducted some preliminary experiments in an effort to extend H-DICE to operate in an off-policy fashion in Appendix C, the core algorithm is still an on-policy algorithm built atop conventional policy-gradient methods. An interesting line of future-work will be to extend credit assignment to off-policy algorithms which are typically more data efficient. By incorporating ideas from OPE and density-ratio estimation to address credit assignment, we hope that other advances in these research areas (Choi et al., 2022) open the door for further symbiotic progress on both fronts.

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

# A  RELATED WORK

The classic workhorse for addressing credit assignment in reinforcement learning has been the eligibility trace (Klopf, 1972; Sutton, 1984; 1988) which awards credit in a geometrically-decaying fashion to states or state-action pairs on the basis of temporal recency to surprising events, as measured by the temporal-difference (TD) error. While these traditional approaches laid the groundwork for the field, the scale of modern problems makes them woefully insufficient and impractical. More recently, a new class of emphatic TD methods have emerged (Sutton et al., 2016; Hallak et al., 2016; Yu, 2015; Anand & Precup, 2021; Chelu et al., 2022) which includes a so-called interest function that enables more nuanced and non-linear attribution of credit for incident states along a sampled trajectory. The theory surrounding these methods, however, operate in general terms without concrete specification of such an interest function. Since scalable methods for leveraging interest functions when learning option initiation sets exist (Sutton et al., 1999b), future work may find it fruitful to learn interest functions using our H-DICE approach in the context of emphatic TD methods. Finally, recent work on expected eligibility traces (van Hasselt et al., 2021; Chelu et al., 2022) acknowledges the trajectory as a random variable and aims to learn eligibility traces that award credit to those realizations of the trajectory that were not observed but could have occurred under the given behavior policy; indeed, the hindsight distribution studied in this work can be shown to emerge through an information-theoretic treatment of credit assignment where such randomness is accounted for (Arumugam et al., 2021). Still, expected eligibility traces suffer from the same temporal recency heuristic that plagues traditional eligibility traces, whereas the hindsight distribution enables non-linear reweighting of on-policy data.

Our work extends the hindsight credit assignment (HCA) work of Harutyunyan et al. (2019), rectifying instabilities in using hindsight distributions for credit assignment. While Alipov et al. (2021) find that an alternative form of clipping applied to hindsight probabilities can improve stability in Atari games, our experiments demonstrate across a broad spectrum of domains that clipping can greatly stall the speed at which learning progresses with HCA. In contrast, an alternative and scalable approach to tackling credit assignment emerges through the concept of return redistribution (Arjona-Medina et al., 2019; Hung et al., 2019; Ren et al., 2022; Gangwani et al., 2020). Unfortunately, practical agents employing return redistribution rely on recurrent network architectures whose performance and susceptibility to vanishing gradients rises as the problem horizon increases. Although methods such as Gangwani et al. (2020) and Ren et al. (2022) do not use recurrent network architectures, they use simplistic methods such as uniform reward decomposition or least squares reward decomposition, which do not necessarily have to assign any meaningful or interpretable credit to the transitions. In contrast, hindsight distributions for data re-weighting retain an elegant idea from classic temporal-difference learning; namely, that local information can be stitched together across timesteps to yield globally effective solutions. There have also been recent approaches that explore the relationship between causality and counterfactuals to the RL credit assignment problem (Mesnard et al., 2021; Buesing et al., 2018); while applying such powerful tools from causal inference does seem like a plausible and even promising avenue for credit assignment in RL, the main drawback to these existing approaches is assuming a perfect causal model of the environment and, with greater limitation, the ability to perform perfect counterfactual inference queries where one factor of variation is manipulated *ceteris paribus* or while keeping everything else fixed. In contrast, H-DICE is theoretically grounded within the framework of importance sampling (Harutyunyan et al., 2019) without such cumbersome assumptions.

## B  Solution to the optimization in Equation 2

In this section, we show that the solution to the following optimization introduced in Section 3.2:

$$\min_{\phi:S \times A \times Z \to \mathcal{C}} \frac{1}{2}\mathbb{E}_{(s,a,z)\sim\mathcal{D}^{h^{\pi_\theta}}}\left[\phi(s,a,z)^2\right] - \mathbb{E}_{\substack{(s,a)\sim d^{\pi_\theta} \\ z\sim\psi(z)}}\left[\phi(s,a,z)\right]$$

is given by

$$\phi^*(s,a,z) = \frac{d^{\pi_\theta}(s,a)\psi(z)}{\mathcal{D}^{h^{\pi_\theta}}(s,a,z)}$$

One can also consult the salient exposition of Nachum et al. (2019) for an alternative rationale and calculation but specialized to their OPE setting instead of our credit assignment focus. We begin by expanding the expectations inside the objective:

$$\frac{1}{2}\mathbb{E}_{(s,a,z)\sim\mathcal{D}^{h^{\pi_\theta}}}\left[\phi(s,a,z)^2\right] - \mathbb{E}_{\substack{(s,a)\sim d^{\pi_\theta} \\ z\sim\psi(z)}}\left[\phi(s,a,z)\right]$$

$$= \frac{1}{2}\int_{s\in S, a\in A, z\in Z}\mathcal{D}^{h^{\pi_\theta}}(s,a,z)\phi(s,a,z)^2 ds da dz - \int_{s\in S, a\in A, z\in Z} d^{\pi_\theta}(s,a)\psi(z)\phi(s,a,z)ds da dz$$

$$= \int_{s\in S, a\in A, z\in Z}\left(\frac{1}{2}\mathcal{D}^{h^{\pi_\theta}}(s,a,z)\phi(s,a,z)^2 - d^{\pi_\theta}(s,a)\psi(z)\phi(s,a,z)\right)ds da dz$$

Minimizing this quantity can be achieved by enforcing first-order optimality conditions. We first differentiate the above integral using the Leibniz Integral Rule, which allows for the derivative to move inside the integral:

$$\int_{s\in S, a\in A, z\in Z}\frac{\partial}{\partial\phi}\left(\frac{1}{2}\mathcal{D}^{h^{\pi_\theta}}(s,a,z)\phi(s,a,z)^2 - d^{\pi_\theta}(s,a)\psi(z)\phi(s,a,z)\right)ds da dz$$

$$= \int_{s\in S, a\in A, z\in Z}\left(\mathcal{D}^{h^{\pi_\theta}}(s,a,z)\phi(s,a,z) - d^{\pi_\theta}(s,a)\psi(z)\right)ds da dz$$

We now equate this expression for the derivative to zero and solve for $\phi$. Similar to the proof in DualDICE Nachum et al. (2019), we look for the value of $\phi(s,a,z)$ such that this gradient is equal to zero pointwise over all state-action-return triples and get:

$$\phi^*(s,a,z) = \frac{d^{\pi_\theta}(s,a)\psi(z)}{\mathcal{D}^{h^{\pi_\theta}}(s,a,z)}$$

which we can then write as:

$$\phi^*(s,a,z) = \frac{d^{\pi_\theta}(s)\pi_\theta(a\mid s)\psi(z)}{\mathcal{D}^{h^{\pi_\theta}}(s)\chi^{\pi_\theta}(z\mid s)h_\omega^{\pi_\theta}(a\mid s,z)}$$

$$\implies \phi^*(s,a,z) = \frac{\pi_\theta(a\mid s)}{\chi^{\pi_\theta}(z\mid s)h_\omega^{\pi_\theta}(a\mid s,z)} \tag{4}$$

The final simplification follows from the fact that $\psi(z)$ is chosen to be a uniform distribution over all possible returns; hence, $\psi(z)$ is a constant term independent of the precise realiziation $z$, therefore,

can be ignored. Ignoring this constant simply changes the magnitude of all policy updates – this is preferred over dividing by the constant, which again, introduces the risk of instability.

For instance, the policy updates are smaller in magnitude when the advantage $A(s, a) \geq 0$ because:

$$\frac{d^{\pi_\theta}(s)\pi_\theta(a \mid s)\psi(z)}{\mathcal{D}^{h^{\pi_\theta}}(s)\chi^{\pi_\theta}(z \mid s)h^{\pi_\theta}_\omega(a \mid s, z)} \leq \frac{d^{\pi_\theta}(s)\pi_\theta(a \mid s)}{\mathcal{D}^{h^{\pi_\theta}}(s)\chi^{\pi_\theta}(z \mid s)h^{\pi_\theta}_\omega(a \mid s, z)}$$

$$\implies \frac{\pi_\theta(a \mid s)\psi(z)}{h^{\pi_\theta}_\omega(a \mid s, z)} \leq \frac{\pi_\theta(a \mid s)}{h^{\pi_\theta}_\omega(a \mid s, z)}$$

$$\implies \left(1 - \frac{\pi_\theta(a \mid s)\psi(z)}{h^{\pi_\theta}_\omega(a \mid s, z)}\right) A(s, a) \geq \left(1 - \frac{\pi_\theta(a \mid s)}{h^{\pi_\theta}_\omega(a \mid s, z)}\right) A(s, a)$$

and vice-versa when $A(s, a) < 0$.

$d^{\pi_\theta}(s) = \mathcal{D}^{h^{\pi_\theta}}(s)$ follows from the fact that $h^{\pi_\theta}$ is the hindsight distribution learned from the state-visitation distribution of $\pi_\theta$. Empirically, we use the same data that we collected from the policy to train the hindsight distribution as well and we update them with the same schedule.

This finally gives us the following expression for the hindsight ratio of interest:

$$\frac{\pi_\theta(a \mid s)}{h^{\pi_\theta}_\omega(a \mid s, z)} = \phi^*(s, a, z)\chi^{\pi_\theta}(z \mid s) \tag{5}$$

There are several choices for $\psi(z)$, which in principle can be any valid distribution over the returns; a good choice of $\psi(z)$ would enable easy computation of the hindsight ratio of interest. There are several potential choices for $\psi(z)$, including $\chi^\pi(z \mid s, a)$ or $\chi^\pi(z \mid s)$. Using the first distribution would require values of $\phi$ to be divided by values of this distribution, risking instability issues that arise when having probabilities or likelihoods as a divisor (indeed, this is the problem we aimed to avoid in the first place). Sampling returns with respect to the latter distribution is especially appealing since it conveniently cancels out with the return predictor distribution which multiples with the output of $\phi$ to obtain the hindsight ratio in Equation 5. However, as shown in Appendix E, sampling returns from $\chi^\pi(z \mid s)$ in the second expectation empirically yields poorer performance than sampling returns from a uniform distribution. We leave other choices for $\psi(z)$ to future work.

Note that we cannot use an arbitrary distribution to model the returns in $\mathcal{D}^{h^{\pi_\theta}}(s, a, z)$ because we need to decompose this term into $\mathcal{D}^{h^{\pi_\theta}}(s)\chi^{\pi_\theta}(z \mid s)h^{\pi_\theta}(a \mid s, z)$. Here, it is important that the returns are sampled from the policy $\pi_\theta$ and that the distribution $\chi^{\pi_\theta}$ is the distribution of returns achieved from a state *under policy* $\pi_\theta$, as this is what the hindsight distribution is conditioned on.

## C  UPDATING THE AUXILIARY MODELS WITH OFF-POLICY SCHEDULES

A potential drawback of H-DICE stems from the fact that four models must be updated at once to learn the hindsight ratio, which can lead to greater training overhead and challenges with hyperparameter tuning. This issue is exacerbated by the fact that all these models must be updated in an on-policy fashion as per the methodology detailed in Section 3.2. This overhead could be alleviated if the auxiliary models could be updated with off-policy schedules, i.e., less frequently than the policy network, thereby allowing them to be trained with data collected from previous versions of the policy. To this end, we aim to study the impact of updating the auxiliary models with various off-policy schedules on final performance.

We perform an ablation on GridWorld-v2 and HalfCheetah environment where we progressively slow down the updates to the auxiliary models relative to the update frequency of the policy. In the plots shown in Figure 4, the curves labeled as $N$x indicate that the auxiliary models are all updated $N$ times slower than the policy model, and as a result with $N$ times more data (which will contain trajectories collected from previous iterations of the policy). Note that auxiliary models' weights are still reset before their updates.

As seen in Figure 4, there is hardly any performance loss when updating with off-policy schedules. In some instances, for example when the auxiliary models are updated 50x less frequently than the policy in HalfCheetah, there see a small improvement over updating all models in lockstep. We conjecture that this similarity in performance may be a result of the trade-off between updating models in a fully on-policy fashion but limiting the volume of training data for the auxiliary models, and being more off-policy which enables models to be trained with more data.

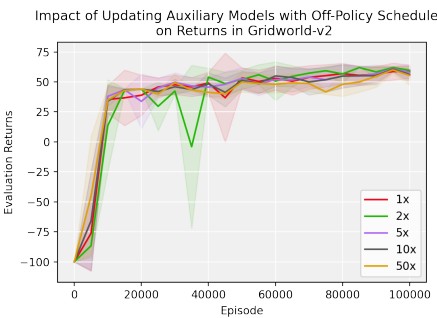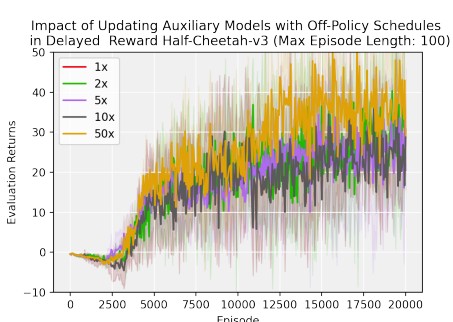

Figure 4: Training curves of varying off policy schedules on GridWorld-v2 and HalfCheetah. There is almost no performance loss when updating with off-policy schedules and in some cases we can even see improvements.

Finally, we note that updating the auxiliary models in an off-policy fashion strays away from the theoretical justification for H-DICE developed in Section 3.2, but enables H-DICE to be computationally efficient with little to no cost to empirical performance.

## D  ABLATION: IMPACT OF HINDSIGHT DICE MODEL RANGE

Figure 5 studies the impact of the output range of the Hindsight DICE model (which is controlled by $C$, as detailed in Section 3.2) on policy performance. All policies are trained with H-DICE, and the only axis of variation across experiments in each environment is the choice of $C$. Results are averaged across 3 seeds for each choice of $C$. Policy performance is similar across choices of $C$ in the Gridworld-v2 and delayed reward Half-Cheetah environments indicating that the choice of this hyperparameter is not very important.

We further note that when $C = 1$, the hindsight ratio calculate by H-DICE lies in the range $[0, 1]$. As a result, the model is not capable of making "double-sided" updates, i.e., down-weighting an action at a state even when the observed returns from that state are positive (and vice versa). This is no longer the case when $C > 1$ but this boost in expressivity risks a potential cost to stability as evidenced by the results of PPO-HCA.

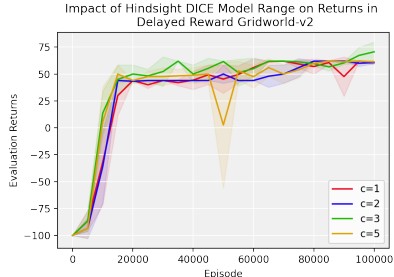 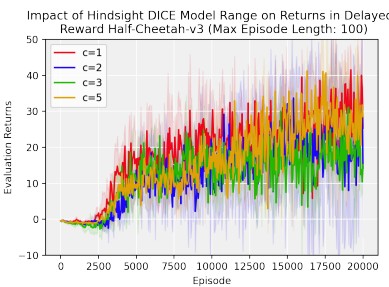

Figure 5: Impact of the range of the Hindsight DICE on policy performance in Gridworld-v2 (left) and Delayed Reward Half-Cheetah with a maximum episode length of 100. For a given value of $C$, the output range of the Hindsight DICE model is set to be $[0, C]$. The performance of H-DICE is similar across choices of $C$.

# E   ABLATION: CHOICE OF $\psi$ RETURN DISTRIBUTION

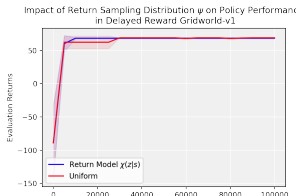 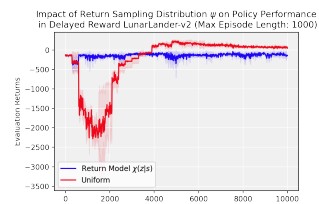 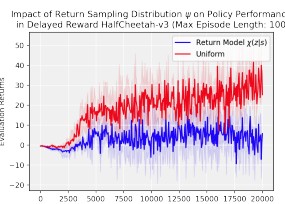

Figure 6: Impact of return sampling distribution $\psi$ on policy performance in Gridworld-v1 and delayed reward LunarLander-v2 and Half-Cheetah-v3. Sampling rewards with a uniform distribution outperforms sampling with respect to the conditional distribution $\chi(z|s)$ in the more difficult LunarLander and HalfCheetah settings.

Figure 6 examines the impact of $\psi$ – the distribution from which returns are sampled in the second expectation in expression 2 – on final performance in Gridworld-v1, LunarLander-v2 with delayed rewards, and Half-Cheetah-v3 with delayed rewards. All policies again are trained with H-DICE, with only the choice of $\psi$ changing between experiments. As justified in Appendix B, we study two options for $\psi$: a uniform distribution over returns, and the state-conditioned return distribution $\chi^{\pi_\theta}(z|s)$. Note that computing the hindsight ratio using the uniform distribution requires $\chi^{\pi_\theta}(z|s)$ to be multiplied with $\phi^*(s, a, z)$, while using $\chi^{\pi_\theta}(z|s)$ enables the hindsight ratio to be directly obtained from $\phi^*(s, a, z)$. Despite both choices being theoretically justified, the results in Figure 6 clearly demonstrate that sampling with the uniform distribution results results in superior performance as compared to sampling from $\chi^{\pi_\theta}(z|s)$. We hypothesise that this is because the Hindsight DICE model alone is not sufficient to capture the complex credit assignment structures in an environment and the scaling offered by the return model is crucial in learning more expressive hindsight ratios. This is further supported by the fact that both choices of $\psi$ perform similarly in the Grid-World but using the state-conditioned return distributions struggles in more complex environments like LunarLander and HalfCheetah.

# F USING H-DICE IN DENSE REWARD SETTINGS

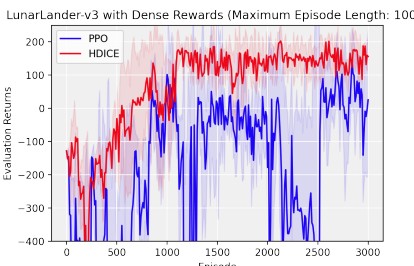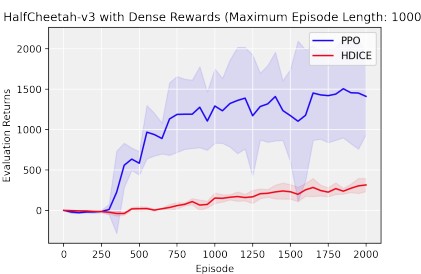

Figure 7: Performance of H-DICE and PPO in LunarLander-v3 and HalfCheetah-v3 (both with maximum episode lengths of 1000) in dense reward settings. Note that these are the default settings of the environments. H-DICE learns stably and quickly in LunarLander, outperforming PPO. However, H-DICE despite learning stably, is much slower than PPO in HalfCheetah.

In Figure 7, we compare the performance of H-DICE with PPO in a dense reward setting, where we don't anticipate a pressing need for effective credit assignment. We seek to answer whether H-DICE can be a competitive substitute to the GAE advantage calculation often used with PPO in a wide range of settings. From the results we can see that H-DICE learns more stably and achieves marginally better performance than PPO on LunarLander over three seeds but is outperformed by PPO on HalfCheetah. We also note that H-DICE was more stable, in terms of variation across seeds, than PPO in both environments.

# G    TRAINING DETAILS AND HYPERPARAMETERS

In this section, we provide training details and hyperparameters for all environments and all methods.

To aid in the stability of training, $\chi_\eta^{\pi_\theta}$, $h_\omega^{\pi_\theta}$, and $\phi_\nu$ utilize techniques such input normalization and clipping of gradient norms. We additionally limit the output range of $\phi_\nu$ by applying a sigmoid at the output to improve training stability (a technique also used by Nachum et al. (2019)), and also find that collecting large amounts of data between updates improved performance.

For training the policy model, we used the usual PPO loss combined with the advantage calculation from 1. To train the Hindsight DICE model, we used the loss from equation 2. To train the hindsight model, we used a cross-entropy loss or a Gaussian log-likelihood loss depending on whether the actions were discrete or continuous. For the return model, we used a Gaussian log-likelihood loss to predict the mean returns from that state and used this distribution to obtain the probability of the given returns.

An extensive list of hyperparameters and the values used for each domain are provided below. While we swept across some of the key hyperparameters such as value loss coefficient, entropy coefficent, number of epochs, etc., other hyperparameters were chosen from Stable Baselines Hill et al. (2018). This is by no means an extensive hyperparameter sweep, especially for the HCA and H-DICE models which have additional hyperparameters as compared to vanilla PPO.

We used PyTorch for our experiments and training was done on single NVIDIA RTX A5000 GPU. The code will be released on acceptance.

## G.1    GRIDWORLD EXPERIMENTS

Table 2: Hyperparameters for the GridWorld experiments for all methods

|  | PPO | PPO-HCA | PPO-HCA-Clip | H-DICE |
|---|---|---|---|---|
| Entropy Coefficient | 0.1 | 0.1 | 0.1 | 0.1 |
| Value Loss Coefficient | 1e-4 | - | - | - |
| PPO Learning Rate | 3e-4 | 3e-4 | 3e-4 | 3e-4 |
| Update Every Episodes | 50 | 50 | 50 | 50 |
| Epsilon Clip | 0.2 | 0.2 | 0.2 | 0.2 |
| PPO Epochs | 30 | 30 | 30 | 30 |
| GAE Lambda | 0.95 | - | - | - |
| Gamma | 0.99 | 0.99 | 0.99 | 0.99 |
| PPO Max Grad Norm | - | - | - | - |
| HCA Max Grad Norm | - | 10 | 10 | 10 |
| Density Pred Max Grad Norm | - | - | - | 10 |
| Return Pred Max Grad Norm | - | - | - | 10 |
| Hindsight Distribution Epochs | - | 10 | 10 | 10 |
| Hindsight DICE Model Epochs | - | - | - | 10 |
| Return Model Epochs | - | - | - | 10 |
| Auxiliary Models Learning Rate | - | 3e-4 | 3e-4 | 3e-4 |
| Auxiliary Models Batchsize | - | 256 | 256 | 256 |
| Return Model Normalize Targets | - | - | - | True |

The PPO policy was initialized with a trunk of 2 hidden layers with 64 ReLU activated hidden units each. The value head and actor head were initialized as separate heads on top of this for PPO. There is no value head for the HCA and H-DICE models.

For the HCA and H-DICE models, each auxiliary model was initialized with 2 layers with 128 ReLU activated hidden units. In the table above, "auxiliary models" refers to the Hindsight Distribution, Hindsight DICE Model and Return predictor together.

## G.2 LUNARLANDER EXPERIMENTS

The hyperparameters that changed from Table 2 are presented below. The rest of the hyperparameters are the same.

Table 3: Hyperparameters for the LunarLander-v2 experiments for all methods

|  | PPO | PPO-HCA | PPO-HCA-Clip | H-DICE |
|---|---|---|---|---|
| Entropy Coefficient | 0.0 | 0.0 | 0.0 | 0.01 |
| Value Loss Coefficient | 0.5 | - | - | - |
| PPO Learning Rate | 3e-4 | 3e-5 | 3e-4 | 3e-4 |
| Update Every Episodes | 300 | 300 | 300 | 300 |
| PPO Epochs | 80 | 80 | 80 | 80 |
| PPO Max Grad Norm | 0.5 | 0.5 | 0.5 | 0.5 |
| Hindsight Epochs | - | 20 | 20 | 20 |
| Hindsight DICE Model Epochs | - | - | - | 1 |
| Return Model Epochs | - | - | - | 20 |

This time, the PPO policy was initialized with a trunk of 3 hidden layers with 128 ReLU activated hidden units each.

For the HCA and H-DICE models, each auxiliary model was initialized with 2 layers with 128 ReLU activated hidden units.

## G.3 HALFCHEETAH EXPERIMENTS

Once again, the hyperparameters that changes from Table 2 are presented below and the other hyperparameters stay the same.

Table 4: Hyperparameters for the HalfCheetah-v3 experiments for all methods

|  | PPO | PPO-HCA | PPO-HCA-Clip | H-DICE |
|---|---|---|---|---|
| Entropy Coefficient | 0.01 | 0.01 | 0.01 | 0.01 |
| Value Loss Coefficient | 0.5 | - | - | - |
| PPO Learning Rate | 3e-4 | 3e-5 | 3e-4 | 3e-4 |
| Update Every Env Steps | 6144 | 6144 | 6144 | 6144 |
| PPO Epochs | 80 | 80 | 80 | 80 |
| PPO Max Grad Norm | 0.5 | 0.5 | 0.5 | 0.5 |

The PPO policy was initialized with a trunk of 3 hidden layers with 128 ReLU activated hidden units each.

For the HCA and H-DICE models, each auxiliary model was initialized with 2 layers with 128 ReLU activated hidden units.

# H  GRIDWORLD-V2

We provide an image of GridWorld-v2 in this section. Note that the dynamics and reward structure of GridWorld-v2 are the same as in GridWorld-v1 as described in Section 4.1 with the exception of the maximum episode length, which is set to 100 in GridWorld-v2 (as opposed to 50 in GridWorld-v1).

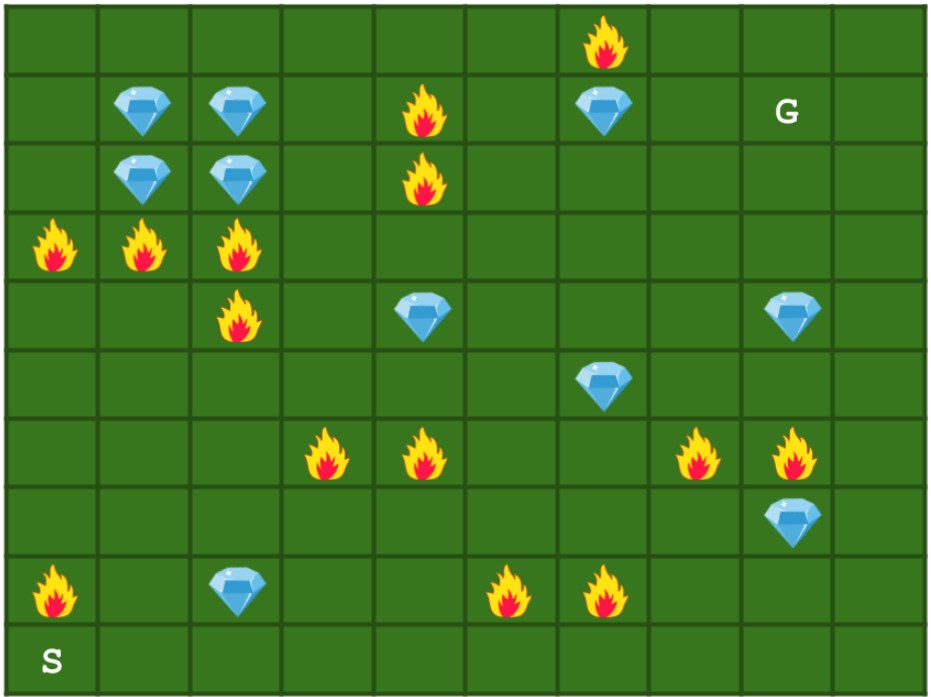

Figure 8: The GridWorld-v2 Environment

