# OpenReview forum: "Hindsight-DICE: Stable Credit Assignment for Deep Reinforcement Learning"
_ICLR.cc/2024/Conference — ICLR 2024 Conference Withdrawn Submission_

### Official Review · Reviewer_UH2T · 2023-10-29

**Soundness:** 3 good
**Presentation:** 3 good
**Contribution:** 3 good
**Rating:** 5
**Confidence:** 4

**Summary:**

This paper addresses the important topic of credit assignment, using a novel approach by drawing inspiration from the OPE literature to improve the stability and therefore the efficiency of HCA. The challenges and solutions are both clearly presented, with empirical results on several simulation tasks.

**Strengths:**

This paper addresses the important topic of credit assignment, using a novel approach by drawing inspiration from the OPE literature to improve the stability and therefore the efficiency of HCA. The challenges and solutions are both clearly presented, with empirical results on several simulation tasks.

Overall, the idea is novel and the problem the authors tried to solve is important. The way the authors present the paper is in general clear.

**Weaknesses:**

The following are my concerns during reading the paper. I'm open to further discussion with the authors and happy to re-evaluate the work if those concerns can be addressed.

### Major:

on the evaluation part:

1. [non-standard settings] Many of the environmental designs are revised in the paper, including the maximal time steps. I wonder if the authors can justify the motivation of using non-standard environments.

2. [baseline] Although the author cited [Zhizhou Ren, Ruihan Guo, Yuan Zhou, and Jian Peng. Learning long-term reward redistribution via randomized return decomposition, 2022.], their method is not included as a baseline.

3. [more comparison] I would expect some larger-scale experiments to see the significance of the improvement and large-scale applicability.
- On the simulation (low action dim control) side, there is a line of work using off-policy methods (which may suffer other stability issues in the context of RLHF but works quite well in Mujoco locomotion)
- On the LM-alignment side, I believe having an experiment on (even a small) language model showing H-DICE improves RLHF will greatly increase the impact of this work.


On the demonstration of results:

I wonder if it is possible to show the learned essential state(or state-actions) with H-DICE? In the demonstrative example in Figure 1, if there are 4 actions at each state, would it be possible to draw a plot visualizing the learned importance in assigning credit to each step?

### Minor:

1. The format file of the submission seems to be changed.

2. 3 seeds are not enough, especially for tasks where statistical significance can not be drawn. e.g., it seems like H-DICE converges very fast in the Humanoid environment.

3. Although in general, the writing is clear, I would recommend the authors to introduce DualDICE more explicitly in the method/preliminary section, to make the method self-contained and easily understandable for readers who might not be familiar with it.

Specifically, \xi is introduced in the main text without explanation; with the help of later context, I understand it as a return model that takes states as input and predicts the trajectory output. However, I’m doubtful if this is a learnable function, as the trajectory return relies on policy. Unlike OPE settings where reward signals are instant, with trajectory-level return the learning of such a return model intuitively will be extremely noisy.

**Questions:**

Please see above.

---

### Official Review · Reviewer_j1vw · 2023-10-30

**Soundness:** 3 good
**Presentation:** 3 good
**Contribution:** 3 good
**Rating:** 6
**Confidence:** 4

**Summary:**

The paper proposes an approach to improve on-policy algorithm performance by leveraging an off-policy evaluation algorithm to improve the stability of hindsight credit assignment by changing the way the hindsight ratios are learned.

**Strengths:**

1. Relative to other areas in RL, there has not been much work since the credit assignment paper in 2019, and its great that this paper revisits the question.
2. I think it is clever to leverage an algorithm from OPE for this alternative use-case.
3. Paper is well-written.
4. The proposed solution is straightforward, and can use off-the-shelf ideas.

**Weaknesses:**

1. While it is great that the proposed algorithm leads to good performance, its not clear why the method actually works/improves upon the existing approach to hindsight ratios. The paper says it is improving "stability", but why does this method actually improve stability? Even if there is an intuition for this, it should be mentioned. To give an example, in OPE we could say DICE methods are more stable than importance sampling methods since they do not multiply a string of ratios as a function of the horizon length.
2. Similar to above, it is unclear why the existing credit assignment ratio computation is unstable.

**Questions:**

1. related to the weaknesses, it is just unclear why the proposed method works. What is an intuition for it?
2. why is the existing credit assignment algorithm/ratio unstable? What leads to its instability?
3. Given the ratio computation is dependent on the dataset used, which leads to bias, is there something that can be said about how much bias there is wrt to optimal ratios and ratios computed by the prior approach?
4. An issue I have noticed with the DICE methods is that sometimes they do not compute the true ratios, but they compute some values that still give reasonable OPE estimates (computing the weighted average of the $\textit{computed}$ ratios is the same as the weighted average of $\textit{true}$ ratios, but the computed ratios themselves are not always equal to the true ratios). I am curious how something like this plays into credit assignment?
5. How much data is actually used when computing all these quantities described in the paper?
6. Few clarifications:
- Section 3.1 says the existing credit assignment algorithm "directly" computes the ratio. This seems to be inaccurate. It appears that $h_{\omega}^{\pi_\theta}$ is computed and then ratio is computed. On the other hand, in the OPE setting, DICE actually does compute the ratio directly instead of numerator/denominator separately and combining them.
- related to above. it seems that the proposed method then also does this indirect computation of $h_{\omega}^{\pi_\theta}$, $\chi_{\eta}^{\pi_\theta}$, solves for $\phi$ using the former two, and then recovers the ratio with $\phi$ and $\mathcal{X}$. Is that correct?

---

### Official Review · Reviewer_dvgs · 2023-11-01

**Soundness:** 3 good
**Presentation:** 3 good
**Contribution:** 3 good
**Rating:** 5
**Confidence:** 4

**Summary:**

This paper aims to address the credit assignment problem in RL, especially the case where the agent only receives a final reward signal after the end of an episode. Technically, this paper proposes H-DICE. It adapts existing importance-sampling ratio estimation techniques for off-policy policy evaluation (OPE) to drastically improve the stability and efficiency of the existing hindsight distribution methods, which were proposed previously for efficient credit assignment in RL. To evaluate the effectiveness of H-DICE, this paper selected GridWorld, LunarLander and MuJoCo Gym Suite, which are representative and didactic scenarios in discrete and continuous control tasks. The main experimental findings are: (1) H-DICE outperforms PPO and HCA variants in the evaluation scenarios; (2) H-DICE gains smaller importance-sampling ratio and thus gets lower variance, which is a core reason for the improved performance of H-DICE compared to PPO-HCA.

**Strengths:**

**Originality:**

This paper adapts an existing importance-sampling ratio estimation technique for the efficient credit assignment in RL. The application is novel and the proposed method is novel. It is interesting to find that the previous OPE method can be utilized to address the credit assignment issue in online RL. To estimate the ratio, supervised learning is applied to estimate the return predictor, hindsight distribution and hindsight DICE model, which is novel.

**Quality:**

This paper has a clear motivation and presents comprehensive related works. The proposed method is novel and solid in quality. Authors conducted massive experiments to demonstrate the merit of the proposed method.

**Clarity:**

Overall, this paper is easy to follow. The abstract is clear and cohesive. This paper presents comprehensive background, related works and formulations. In the methodology section, it is easy to follow and understand the key technique contribution. The evaluation results are convincing.

**Significance:**

The credit problem on which this problem focus is significant. It brings insights to efficient credit assignment in RL.

**Weaknesses:**

The key weaknesses of this paper are:

1. The application of the existing DualDICE in credit assignment is straightforward and simple. It would be better to include technical motivation and more understanding on why it can be applied.
2. It is not easy to follow Equation (2) in the approach section. Why is the Equation (2) feasible?
3. Most of the evaluation scenarios are simple and didactic. It would be great to see results on large scale experiments on some Atari games and small-scaled Go, where the reward is often delayed and shown in the end of an episode.
4. The presentation of Section 4.5 is not very good. It is not straightforward to just pick one state as an example case. You can use heatmap to visualise the result of a trajectory. You mentioned the variance was reduced, so you can also plot the finding to convince readers.

**Questions:**

Q1: Supervised learning adds additional computational time complexity in RL. Could you please analyse it?

Q2: Can you introduce the hyperparameters, such as the batch size, learning rate, the number of epochs, used in supervised learning?

---

### Official Review · Reviewer_L6UQ · 2023-11-08

**Soundness:** 1 poor
**Presentation:** 3 good
**Contribution:** 2 fair
**Rating:** 1
**Confidence:** 5

**Summary:**

The authors address credit assignment in deep reinforcement learning to improve sample efficiency. They propose Hindsight Distribution Correction Estimation (H-DICE), a variant of Hindsight Credit Assignment (HCA) that attempts to mitigate the instability of importance-sampling ratios in the distribution correction. The method is evaluated using PPO in a new GridWorld, LunarLander, and several MuJoCo environments where reward information is forcibly delayed until the end of the episode. The tested baselines are PPO and other variations of it that use HCA.

**Strengths:**

- Overall, the paper is well organized and easy to follow.
- The references in the paper are relatively thorough and shows research effort from the authors. However, I feel that the discussion misses a number of important references related to credit assignment and overstates the novelty of the work (see weaknesses below). The related work section is currently relegated to the appendix, but this should be featured in the main paper in my opinion.
- Other than the violation of the Markov reward property (see weaknesses below), the authors consider a diverse range of environments and follow reasonable experimental practices. I think there is potential for a publication in the future if the authors are able to more fairly evaluate their method in these tasks. I like the new GridWorld environment as a focused credit-assignment study, and perhaps it could be modified to demonstrate the issues tackled by the authors in a more principled way. Unfortunately, the current empirical results do not support the hypotheses or claims made by the authors, as I discuss below.

**Weaknesses:**

- The main contribution is incremental, essentially applying an existing technique from DualDICE to the existing framework of HCA. There are no new theoretical results or analysis of how the proposed method mitigates the purported instability of HCA.
- The experiment results are undermined by the “delayed-reward” setting introduced by the authors. During an episode, rewards are accumulated, and the final sum is presented to the agent only upon episode termination. This makes the reward function extremely non-Markov, which the authors acknowledge (Section 4.1). However, this is a major issue. All of the baselines (which are just PPO with or without various forms of HCA) rely on a value function assuming Markov states; it is impossible to learn $V(s)$ in this partially observable setting, since the agent does not have access to the history of observations. Notably, this is not a matter of credit assignment; the given baselines are formulated based on the Markov assumption that does not hold in the experiments. Without this assumption, the value function is not well defined. There is simply no way for these methods to learn $v_\pi$ or $v_*$, even with an infinite amount of data—the agent does not have the information available to make the problem learnable. On the other hand, H-DICE does not learn a value function, and I suspect this is the main reason it is able to outperform the baselines. Despite this, the scores achieved in the MuJoCo environments are low compared to the standard benchmark, making me question how well the H-DICE agent is actually learning these tasks and whether the authors’ goal of a “demonstrably stable” method has really been achieved. I would like to see how H-DICE performs in comparison to PPO on the standard MuJoCo benchmark or other Markov-reward environments. If the authors’ method is as successful at credit assignment as they claim, then it should be able to outperform PPO in these environments, too.
- Some claims about credit assignment in reinforcement learning are incorrect. The authors say that PPO with GAE (i.e., $\lambda$-returns) “does not perform explicit credit assignment.” This is simply not true. The $\lambda$-return very explicitly assigns credit on the basis of temporal recency, with less credit given to states far into the future. In fact, all deep RL methods perform credit assignment to some extent (usually through the value function itself), because if they did not, they would simply be unable to learn anything. Perhaps what is meant instead is that these methods are assigning credit only based on temporal information and not more sophisticated techniques. Still, there are many papers about more advanced credit-assignment techniques for deep RL (e.g., [1,2,3] and the cited HCA paper) despite the authors’ assertion that that there has been “relatively little consideration given to issues of credit assignment.” See the references of these papers for other related works.
- PPO is incorrectly described as an off-policy method, but in reality it learns from on-policy experience. The importance-sampling ratio in the objective comes from the sample approximation in TRPO [4]. The clipping is applied by PPO to generate a surrogate objective that aims to simulate the trust-region effect.

**References**

[1] Safe and efficient off-policy reinforcement learning. Remi Munos, Tom Stepleton, Anna Harutyunyan, Marc G. Bellemare. 2016.

[2] Hindsight experience replay. Marcin Andrychowicz, Filip Wolski, Alex Ray, Jonas Schneider, Rachel Fong, Peter Welinder, Bob McGrew, Josh Tobin, Pieter Abbeel, Wojciech Zaremba. 2017.

[3] Expected Eligibility Traces. Hado van Hasselt, Sephora Madjiheurem, Matteo Hessel, David Silver, Andre Barreto, Diana Borsa. 2021.

[4] Trust region policy optimization. John Schulman, Sergey Levine, Philipp Moritz, Michael Jordan, Pieter Abbeel. 2015.

**Questions:**

What are the input features for the GridWorld? Are you still using neural networks for this problem?